# Maximum Run-Up and Alongshore Mass Transport Due to Edge Waves

**Ho-Jun Yoo** [1], **Hyoseob Kim** [2], **Changhwan Jang** [3,*], **Ki-Hyun Kim** [1] **and Tae-Soon Kang** [1]

1   Department of Coastal Management, Geosystem Research Corp, Gunpo 15807, Korea;
    yoohj@geosr.com (H.-J.Y.); khkim@geosr.com (K.-H.K.); kangts@geosr.com (T.-S.K.)
2   Department of Civil Engineering, Kookmin University, Seoul 02707, Korea; hkim@kookmin.ac.kr
3   Division of Smart Construction and Environmental Engineering, Daejin University, Pocheon 11159, Korea
*   Correspondence: cjang@daejin.ac.kr

**Abstract:** The edge wave on a uniform-sloped seabed was described by the velocity-potential function by Mok and Yeh in 1999. Edge waves cannot be extended above a certain level from the still-water level, and the upper limit of the run-up of the edge waves for given conditions is found here. In this study, quantitative mass transport by the edge waves of the beach is introduced. The maximum run-up height is decided from the wave's amplitude at shoreline, and the maximum run-up distance from the shoreline is proportional to the wavelength of the edge waves. The fluid alongshore-mass-transport profile shows that the strongest mass transport rate corresponds to the position offshoreward multiplied by 0.0362 times the wavelength, and its magnitude is 1.23 times the mass-transport rate at the shoreline. The maximum cross-sectional total mass-transport rate is 0.214 times the mass transport at the shoreline, multiplied by the wavelength for the maximum run-up condition. This study suggests that edge waves cannot be increased infinitely and that there is a maximum run-up on the coast.

**Keywords:** edge waves; wave run-up; maximum edge waves; beach erosion

## 1. Introduction

Coastal morphological changes have interested many researchers in the fields of coastal engineering and oceanography. Morphological changes, such as beach scarps, directly affect the safety of artificial structures around dunes, while underwater morphological changes affect seabed ecology [1,2], maintenance of navigation channels [3], and the sustainability and resilience of beaches [4–7].

Scarps are closely linked to wave run-up. Wave run-up is closely related to beach cusps, and beach cusps may be exposed parts of rhythmic, underwater bedforms near the shoreline and the surf-zone terrace. Shore-normal wave-induced run-up has been treated as the highly nonlinear deformation of waves hitting the beach in a nearly normal direction. However, edge waves also have run-up. We need to know the order of the edge wave-induced run-up magnitude at fields. Some waves travel along shoreline at the edges of coasts, and they are called "edge waves".

Edge waves, which have long been known as curiosities by researchers such as Lamb [8], have recently received much attention as a research target. This is due to the fact that edge waves clearly play an important role in the dynamics of coastal areas and beach-erosion processes [9]. Unfortunately, field edge waves and their run-up have not yet been reported well. Possible driving forces of edge waves may include low-pressure movement along coastline [10], under-sea earthquakes [9,11], or incident waves or their broken waves approaching almost normally near the coastline [12–14]. Existing theories on the alongshore wave-induced current, driven by radiation-stress gradients of obliquely approaching, short-period waves describe the underwater flow pattern [15]. However, the steady alongshore current field induced by wave-average properties cannot describe

the flow above the wave set-up level. Wave-resolving theories exist, and they are keen to breaking criteria, and the driven, alongshore current develops mainly around the breaker line, not around the beachface.

The concept of edge-wave development being due to normally approaching waves includes the wave reflection at the shoreline and refraction of the reflected waves, but this concept has not been mathematically explained yet [16–18]. We know that edge waves do exist. They have been generated in three-dimensional laboratory wave basins. For example, edge waves have been generated with a vertical pedal with a hinge [19–21]. Some questions still remain regarding edge wave attributes. First, it has not been shown whether they can maintain their shape while travelling a long distance. Edge waves have travelled only a few wavelengths due to the limited laboratory basin space available for Mok and Yeh's experiments [21]. Second, the magnitude of edge waves at coasts has not been evaluated. If edge waves are high, the importance of their existence also becomes high. Beach cusps and rhythmic bedforms along coastal edges have often been observed along straight beaches [22,23], and some researchers argue that they are closely linked to edge waves, based on the reasoning that the periodic bedform lengths should match the edge waves' lengths. However, the above assumption has not been proven yet.

When edge waves with a given wave period are in a resonance mode, the water surface will preserve the wavelength, wave crest lines or trough lines will be stationary, and therefore, the bedform lengths may be assumed to be a multiple of the wavelength in the alongshore direction. In reality, the sediment particle movement is much complicated, including saltation, suspending, and settling, and the relationship between the spectra of the bedform lengths and the wavelengths of the edge waves should be carefully examined.

Difficulties in separating out edge waves from complex, multi-superposed field waves with a spectrum may come from various causes: infiltration at the bed boundary, viscosity in the bed-boundary layer, turbulence, or highly nonlinear behavior at the water's surface, such as waves breaking. Some of the above difficulties may exist even in well-controlled laboratory experiments. Due to the lack of a satisfactory confirmation of the existence of edge waves from field data, the importance of edge waves on coastal morphological shaping has not yet been addressed well. We could say that even, if edge waves are not high, they may effect morphological change slowly but persistently, influencing the formation of cusps, bedforms, or run-up, and contributing to the development of scarps between dunes and beachfaces. It is obvious that edge waves should be treated more seriously since, if their magnitudes at fields are significant, they can contribute to beach morphology profoundly.

Edge waves have been described as mathematical theories since 1940. Governing equations of coastal edge waves are virtually the same as the three-dimensional continuity equation and the three momentum equations, i.e., the Navier–Stokes equations. There have been two distinctly different approaches: shallow water waves and irrotational potential-function waves.

The final equations for water levels are slightly different from each other due to the different interim assumptions. Nonlinear solutions are not available yet, weakly nonlinear solutions have been proposed by some researchers [24–28], and linear solutions have been well-introduced from both the shallow-water assumption and the velocity-potential functions [10,29,30]. Most existing edge-wave formations are based on the separation of variables concept.

The mass transport due to edge waves has been described by Weber and Ghaffari [9]. However, their theory does not cover the on-shore zone. Mok and Yeh [21] carried out laboratory experiments and measured mass transport due to edge waves. However, they focused on the vertical distribution of the mass transport in underwater zones.

We have adopted an existing solution set rooted in the potential function assumption for progressive edge waves proposed by Mok and Yeh [21]. We focus on the horizontal–vertical distribution of mass transport due to edge waves underwater and on the beach. As a first step, progressive edge waves propagating parallelly along the shoreline or the bed-level contour lines are considered here. In other words, edge waves are not oblique to

the shoreline. Two opposite edge waves can form a resonance pattern, but the resonance of the edge waves is not treated here. Section 2 proposes the maximum run-up limit formula. Section 3 introduces the possible alongshore mass transport in the coastal field scale using the maximum run-up limit formula and quantitatively analyzes the results. Then, the conclusion of this paper is given in the Section 4.

## 2. Finding the Maximum Run-Up Limit

The formulation of equations to describe edge waves includes governing equations and boundary conditions. The onshore boundary condition is especially important as it defines the characteristics of an edge wave. Any kind of folding of the bed boundary, such as that shown in Figure 1a,b deteriorates the behavior of an edge wave. We focus on waves over a straight profile with a uniform slope, as shown in Figure 1c, where the onshore boundary limit meets the bed boundary.

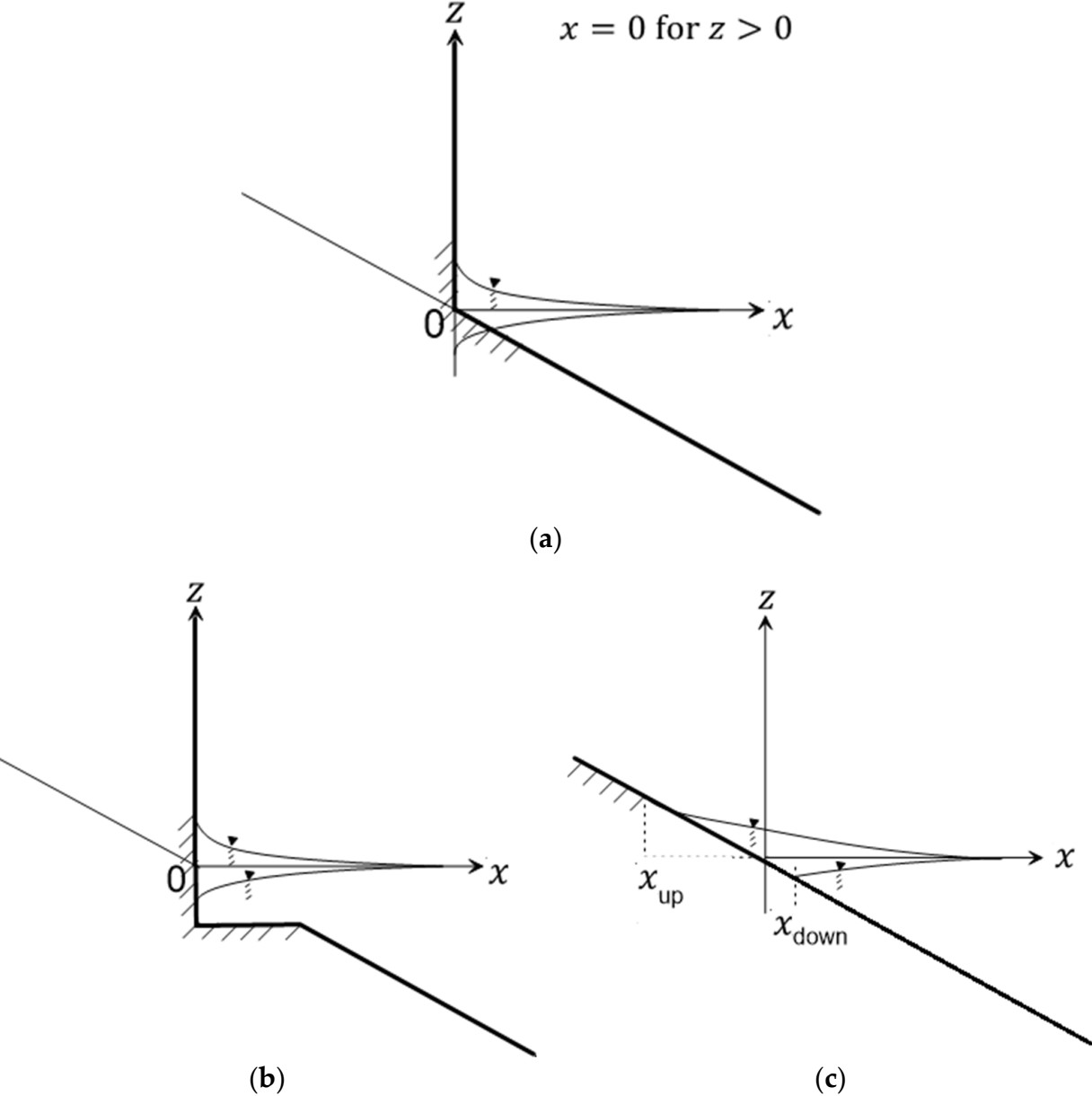

**Figure 1.** Folded, excavated, or sloped bed boundaries for edge waves: (**a**) vertical wall at shoreline; (**b**) excavation around shoreline; (**c**) uniformly sloped bed.

We use convenient, inclined coordinates $(x_s, y, z_s)$ for the velocity-potential function, and we use the horizontal and vertical coordinates $(x, y, z)$ for the displacement of the water surface from the still-water level, as shown in Figure 2.

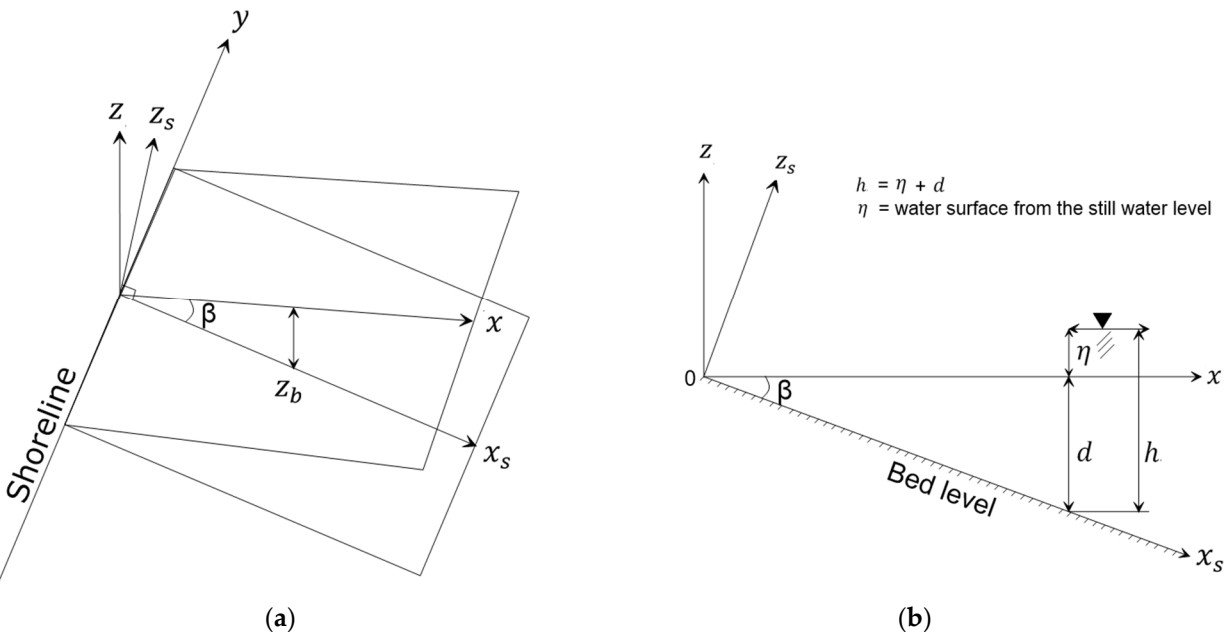

**Figure 2.** Description of the two coordinates: (**a**) three-dimensional view; (**b**) $x - z$ section.

The governing equation of an edge wave is the Laplace equation derived from the fluid mass conservation.

$$\nabla^2 \phi_s = 0 \tag{1}$$

$$V_s = \nabla \phi_s \tag{2}$$

where $\phi_s$ is the velocity-potential function, $V_s$ is the velocity vector $(u_s, v, w_s)$, and $u_s, v, w_s$ are the fluid velocity components in the $x_s, y, z_s$ directions, respectively.

Mathematical solutions cover the whole domain, including the upper zone above the bed and the lower zone below the bed, but only a partial portion of the solutions is valid from the physical point of view. See Figure 3.

The kinematic boundary condition on the water's surface is:

$$\omega_s = \frac{D\eta}{dt} \qquad on \quad z = \eta \tag{3}$$

The kinematic boundary condition on the seabed is:

$$\omega_s = 0 \ on \ z_s = 0 \ (z = -x \tan \beta) \tag{4}$$

where $\beta$ is the angle between the horizon and the seabed, and $\eta$ is the displacement of the water's surface from the still-water level.

The dynamic free-surface boundary condition is:

$$\frac{p}{\rho} = 0 \ on \ z = \eta \tag{5}$$

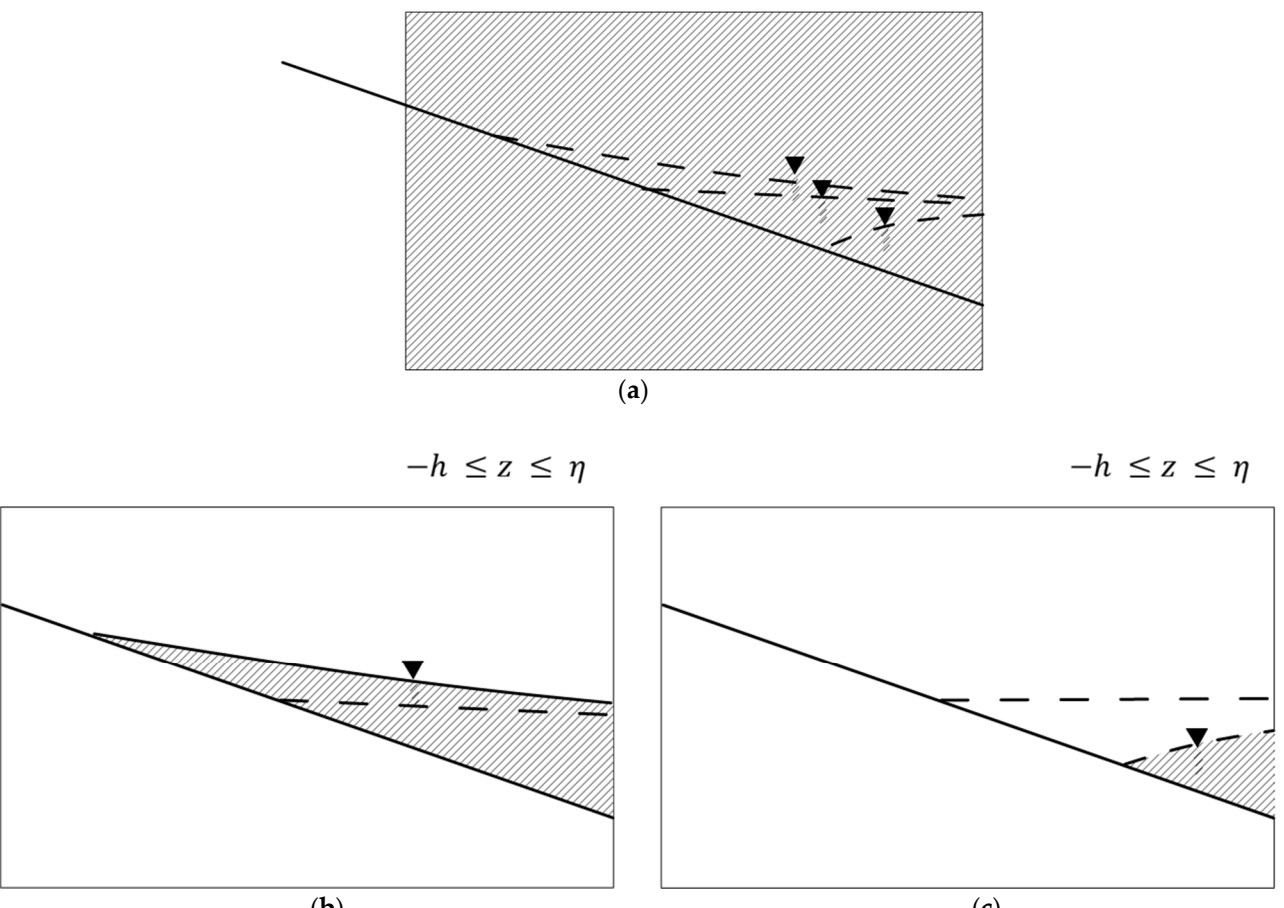

**Figure 3.** Physically valid domain vs. mathematical domain: (**a**) mathematical domain: whole domain; (**b**) valid between crest and bed; (**c**) valid between trough and bed.

where $p$ is the fluid pressure and $\rho$ is the fluid density. Analytical solutions have been proposed by Mok and Yeh [21]. Taking Ursell's mode 0 from their solutions [31], we have the potential function, the surface displacement, and the dispersion relationship, which are:

$$\phi_s(x_s, y, z_s, t) = \frac{\alpha g}{\omega} \exp(-kx_s) \sin(ky - \omega t) \tag{6}$$

$$\eta(x, y, t) = \alpha \exp(-kx \cos \beta) \cos(ky - \omega t) \tag{7}$$

$$\omega^2 = gk \sin \beta \tag{8}$$

$$u_s = -\frac{\alpha g k}{\omega} \exp(-kx_s) \sin(ky - \omega t) \tag{9}$$

$$v = \frac{\alpha g k}{\omega} \exp(-kx_s) \cos(ky - \omega t) \tag{10}$$

$$w_s = 0 \tag{11}$$

where $\alpha$ is the wave amplitude at the shoreline ($x = 0$), $g$ is the acceleration due to gravity, $k$ is the wave number, $\phi_s$ is the velocity potential of Ursell's mode 0, $\beta$ is the beach slope angle, $\omega$ is the wave angular frequency, and $t$ is time.

The run-up and run-down propagate along the shoreline with a constant celerity. The above solution for the displacement of the surface level limits the run-up which is a multiplication of an exponential function and a sinusoidal function. See Figure 4.

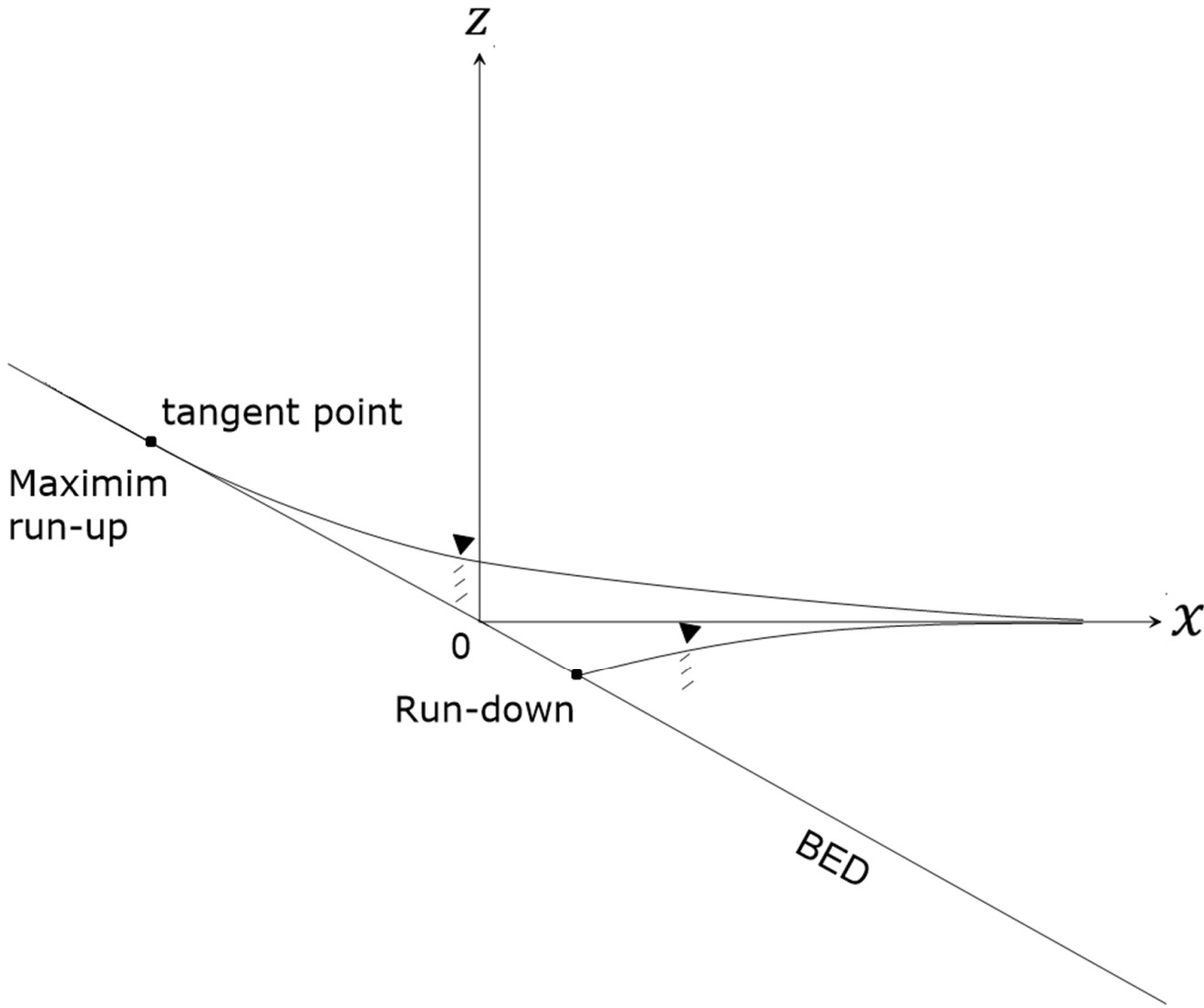

**Figure 4.** Maximum possible run-up of an edge wave.

The upper envelope of the water's surface level corresponds to the case when the wave phase is $0°$.

$$ky - \omega t = 0 \eta(x, 0, 0) = \alpha \ \exp(-kx \cos \beta). \tag{12}$$

The water's surface makes a tangent to the seabed line for the maximum run-up. The seabed is expressed as:

$$z_b(x) = -x \tan \beta \tag{13}$$

As the bed level line is tangential to the water's surface curve, we apply the following two equations: Equating Equations (12) and (13),

$$\alpha \ \exp(-kx \cos \beta) = -x \tan \beta \tag{14}$$

For another example, derivatives of Equations (12) and (13) should match, as:

$$\frac{d\eta}{dx} = \frac{dz_b}{dx} \tag{15}$$

Then, we find the position of the maximum run-up:

$$x = -\frac{1}{k\cos\beta} \tag{16}$$

Accordingly, the maximum run-up height becomes

$$\eta = e\alpha_{max} \tag{17}$$

Inserting the above relationship into Equation (13):

$$\alpha_{max} = \frac{\tan\beta}{ek\cos\beta} \tag{18}$$

When the water's surface amplitude at the shoreline $\alpha$ is known, the wave number $k$ is decided if maximum run-up develops, or vice versa. The water's surface maximum amplitude at the shoreline is $\alpha_{max}$; the base of the natural logarithm is $e$. The instantaneous water depth $h$ is the sum of the still-water depth $x\tan\beta$ and the surface displacement from the still-water level $\eta$:

$$h = \alpha\exp(-kx\cos\beta)\cos(ky - \omega t) + x\tan\beta \tag{19}$$

Assuming $s$ mild-slope seabed, the fluid flow velocity in the $y$ direction is:

$$v = \frac{\alpha g k}{\omega}\exp(-kx\cos\theta)\sin(ky - \omega t) \tag{20}$$

The wave-period-average fluid flux in the $x$ direction is zero.

$$q_x = \frac{1}{T}\int_0^T hu dt = 0 \tag{21}$$

$$Q_x = \int q_x dx = 0 \tag{22}$$

where $q_x$ is the mass transport for a specific position, $Q_x$ is an on-offshore integration of $q_x$ over the entire section.

The wave-period-average fluid flux in the $y$ direction $q_y$ is a function of $x$:

$$q_y = \frac{1}{T}\int_0^T hv dt \tag{23}$$

At $x = 0$, we have an approximate solution for $q_y$,

$$q_y(0) = \frac{1}{4}\frac{g\alpha^2 k}{\omega} \tag{24}$$

The $y$-directional total fluid flux across a section in the $x$ direction is obtained from the integration of $q_y$ in the $x$ direction:

$$Q_y = \int q_y dx \tag{25}$$

We introduce a non-dimensional integration coefficient $\delta$ as:

$$Q_y = \delta q_y(0)L \tag{26}$$

The $\delta$ can be a stable and universal coefficient of 0.214 if the edge waves satisfy the maximum run-up condition. However, if the edge waves are lower than the maximum run-up condition, the integration coefficient of $\delta$ becomes smaller, as shown in Figure 5.

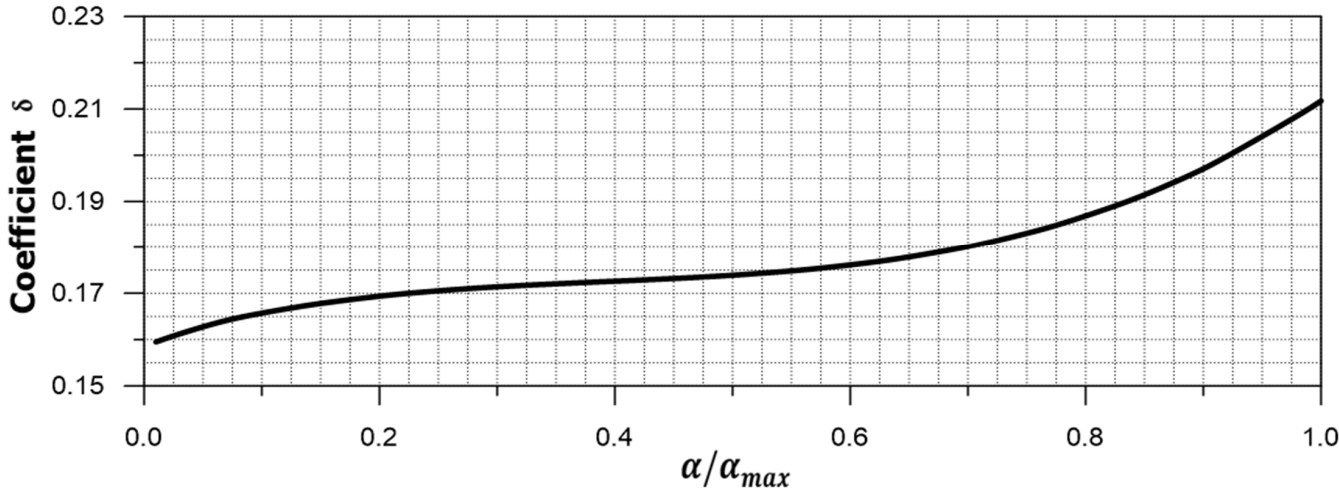

**Figure 5.** Coefficient $\delta$ versus $\alpha/\alpha_{max}$.

## 3. Field Scales of Alongshore Mass Transport

We examined the magnitude of the mass transport profile at real coastal fields. In order to analyze the distribution of mass transport along the coast, mass transport was analyzed as a test case of real fields. We took two cases of wave conditions which are typical on the East Coast of Korea. See Table 1.

**Table 1.** Test case to examine the magnitude of mass transport.

| Case | $\omega(s^{-1})$ | $\mathcal{L}(m)$ * | $k(m^{-1})$ | $\alpha(m)$ | $\tan\beta$ (deg.) | Remark |
|------|------------------|--------------------|-------------|-------------|---------------------|--------|
| 1 | 0.579 | 62.8 | 0.100 | 1.43 | 20 | $\alpha_{max}$ |
| 2 | 0.579 | 62.8 | 0.100 | 0.71 | 20 | $\alpha_{max}/2$ |

*: wave length.

The flow fields on the water's surface of the edge waves are shown in Figure 6, which describes the overall propagation of the edge waves in the $y$ direction. The alongshore fluid flux distributed two sections in the $x$ direction, as shown in Figure 7. Point A shows that $q_y/q_y(0)$ is 1.00(-), which is thought to be significant, considering the surrounding shallow bathymetry. Point B shows that $q_y/q_y(0)$ is 1.23(-), nondimensional offshore distance from shore $(x/\mathcal{L})$ is 0.0362(-). The total sectional fluid flux is computed as 11.2 m$^3$/s. It is obvious that the mass transport plays an important role in the transporting of sediment at the beachface. If the wave height is less than that for the maximum run-up case, the alongshore mass transport rate will be reduced, and the resultant total sectional transport rate will be reduced. Talking the example of Case 2, the total sectional fluid mass transport rate is 2.37 m$^3$/s, which is about a quarter of that for the maximum run-up situation.

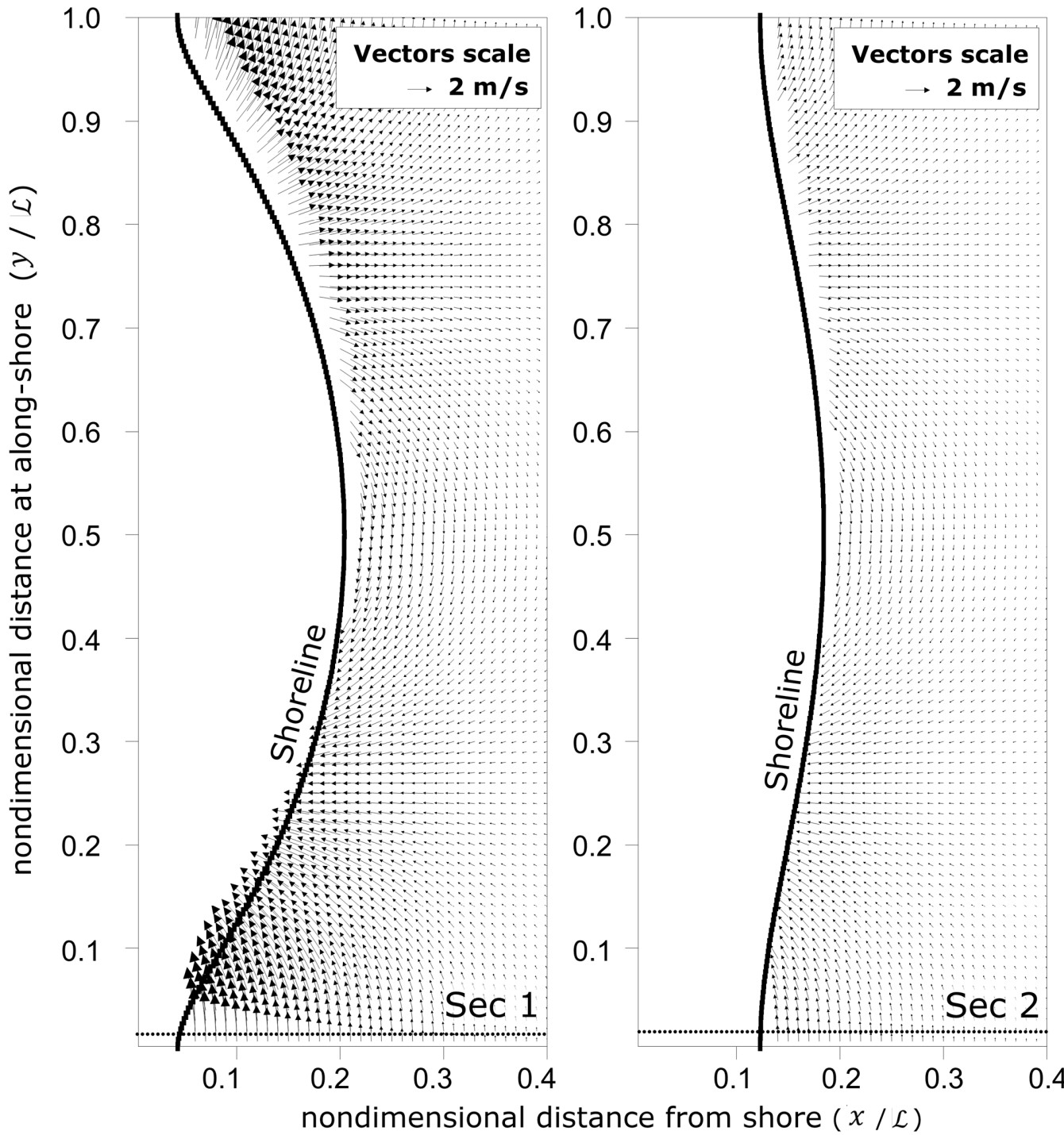

**Figure 6.** Flow field of progressive edge waves in the y axis: (**left**: Case 1, **right**: Case 2).

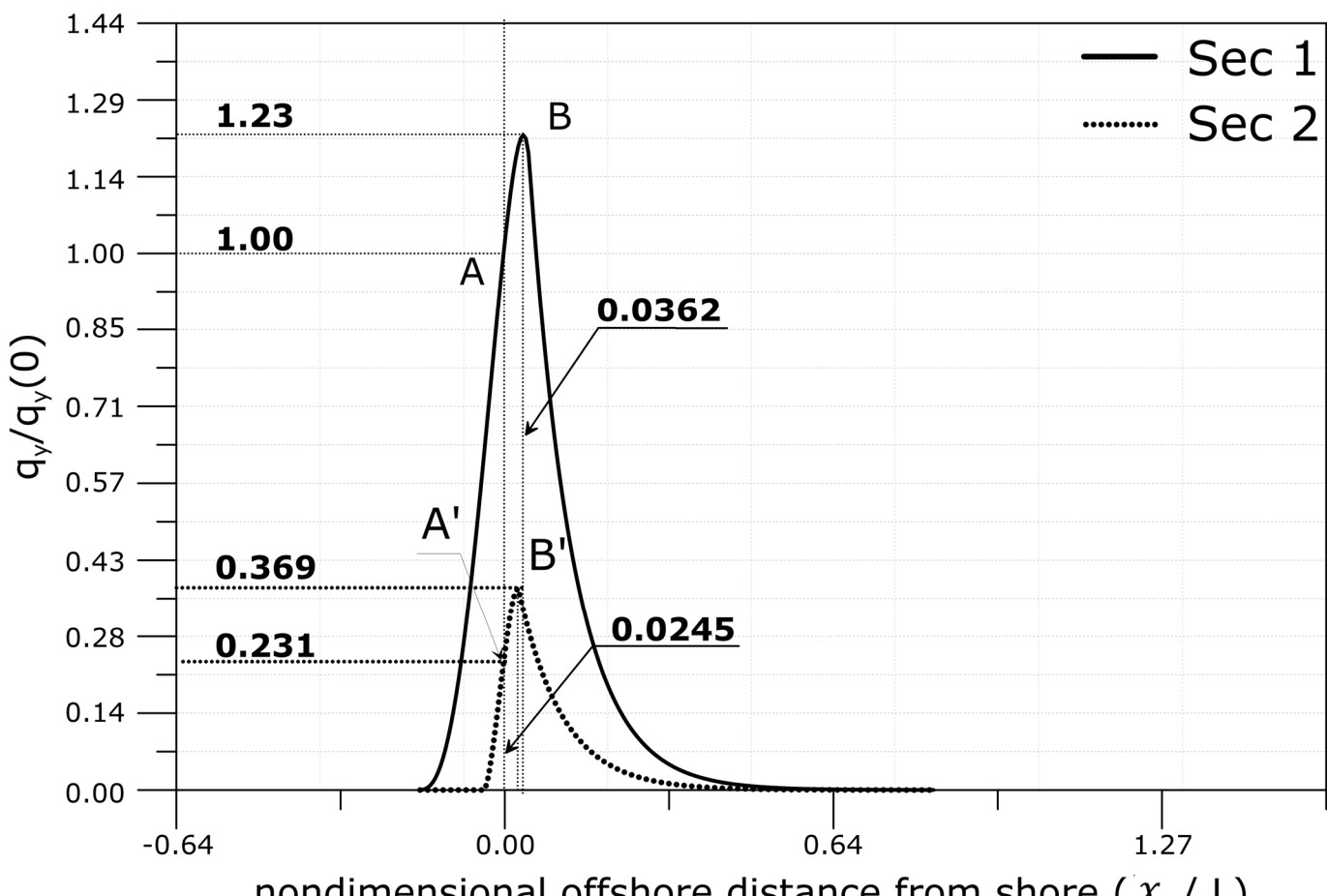

**Figure 7.** Alongshore fluid mass transport distribution.

### 4. Conclusions

Little has been reported on alongshore sediment transport around beachfaces due to both the difficulty in measurements and the inherent mechanical complexity of flow and sediment movement there. The movement of sediments in the swash zone and the change in the shape of the seabed are actually very complicated issues. The slope of the beach varies depending on the wave and flow conditions. The bedform is mainly due to the action of waves, changes in drag due to bed roughness, the introduction of shape drag, and the generation of energy loss due to the secondary current. The problem is further complicated by sediment movement due to the mixture of water and sediments in the swash zone. In this problem, the boundary layer's properties may not behave like clear water, which makes accurate decisions much more difficult.. The present mass transport due to edge waves is different from the obliquely approaching wave-induced alongshore current in the sense that the two currents cover different zones. See Figure 8.

Edge waves over a constant-sloped seabed include run-up and alongshore mass transport. Conventional potential function theory can be applied to edge waves to find the maximum possible runup position and height.

If we applied the existing potential function theory to edge waves, we can found the the possible maximum run-up position and height. Approximate alongshore mass transport distribution in the offshore direction was computed, and the result shows that the maximum alongshore mass transport occurs at 0.0362 times the edge wave's wavelength from the shoreline. The maximum mass transport rate $q_y/q_y(0)$ is 1.23 times larger than that at the shoreline.

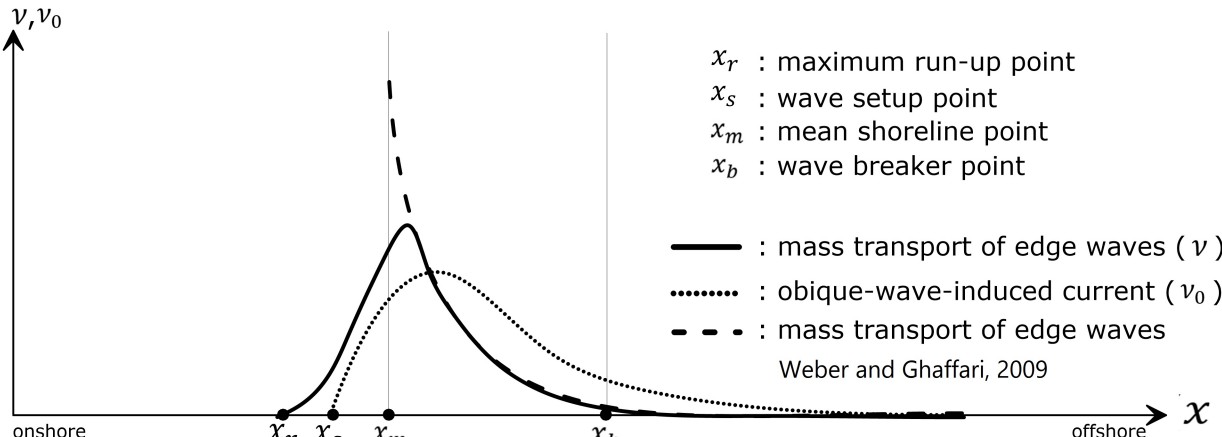

**Figure 8.** Concept of edge wave mass transport and oblique wave-induced current distribution [9].

The total fluid flux, or the integrated alongshore mass transport, the coefficient $\delta$ was found to be 0.214 the mass transport rate at the shoreline and the wavelength at the maximum run-up conditions. The coefficient $\delta$ was found to reduce to 0.159 as the ratio between the wave amplitude at shoreline and the wavelength for the maximum run-up situation decreases.

The present computational results are based on the ideal fluid assumption. The total mass transport around the beachface implies strong sediment transport in the zone. However, the real flows on the coast involve bed friction, infiltration through sands or gravels, and laminar or turbulent bed-boundary-layer formation. The sediment transport involves more factors on top of those. The possibility of the existence of edge-waves involving various mechanisms and how to calculate the mass transport needs more follow-up studies. Also, further research could link the present fluid mass transport due to the edge waves to sediment transport, morphological change onshore, and the sustainability of beaches.

**Author Contributions:** Conceptualization, H.K. and H.-J.Y.; methodology, H.K. and H.-J.Y.; validation, C.J., K.-H.K. and T.-S.K.; formal analysis, H.K. and H.-J.Y.; data curation, C.J., K.-H.K. and T.-S.K.; writing—original draft preparation, H.K. and H.-J.Y.; writing—review and editing, H.K. and H.-J.Y.; visualization, H.K. and H.-J.Y.; supervision and project administration, H.K.; funding acquisition, C.J. All authors have read and agreed to the published version of the manuscript.

**Funding:** This work was supported by a Daejin University Research Grant in 2022.

**Informed Consent Statement:** Informed consent was obtained from all subjects involved in the study.

**Data Availability Statement:** The datasets used and analyzed in the current study are available from the corresponding author upon reasonable request.

**Conflicts of Interest:** The authors declare no conflict of interest.

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
