# Peer review of "Maximum Run-Up and Alongshore Mass Transport Due to Edge Waves"

_jmse, doi:10.3390/jmse10070894_

Round 1
Reviewer 1 Report
The paper utilized the velocity potential function to determine the maximum run-up height and alongshore fluid mass transport rate of edge waves. The work is helpful for better understanding of coastal hydrodynamics of edge waves.
I suggest a major revision on the manuscript before it can be accepted for publication.
(1) Figure 2(b) should be improved with η clearly shown and the x_s and z_s included. Besides, the η to show the water surface is not easy to understand.
(2) Figure 3 as well as the paragraph (Lines 114 – 116) seems not necessary. The meaning is not clear.
(3) What is Eq. (14) used for? According to my understanding, in the paper, the run-up limit of edge waves is determined by Eq. (15), the gradient of the wave surface height, and Eq. (14) is not used and explained in the following texts. In my mind, the run-up limit should be determined by Eq. (14) rather than Eq. (15). Please give some justifications.
(4) Eq. (17), what is e? The meaning of notation should be offered.
(5) What is alpha_max? The meaning should be offered.
(6) Line 164, how is the value ‘0.214’ obtained?
(7) The writing and language should be improved. For exmaple,
(7.1) Line 39, ‘have’ should be ‘has’;
(7.2) Line 95, ‘Figure 1(a) of Figure 1(b)’ may be ‘Figure 1(a) and 1(b)’
(7.3) Line 173, ‘ Two example cases are used to examine could demonstrated ... ‘. Error in grammar.
(7.4) More ...
Author Response
Point 1: The paper utilized the velocity potential function to determine the maximum run-up height and alongshore fluid mass transport rate of edge waves. The work is helpful for better understanding of coastal hydrodynamics of edge waves.
Response 1: Thank you for the reviewer's sincere response. Previous studies focused on underwater edge-wave. This paper is hoped to contribute some academic significance in quantitatively analyzing the possibility and size of edge wave in the beach.
The author's response to reviewer`s comments are summarized below. Please take this into consideration in the review.
Point 2: Figure 2(b) should be improved with η clearly shown and the x_s and z_s included. Besides, the η to show the water surface is not easy to understand.
Response 2: In Figure 2, the meaning of each symbols were modified to be elaborately specified in the text and the picture. , are inclined coordinates for the velocity potential function.
Point 3: Figure 3 as well as the paragraph (Lines 114 – 116) seems not necessary. The meaning is not clear.
Response 3: Figure 3(b) and 3(c). are showing that partial portions of the solutions are valid area from the physical point of view. In the sub-title of Figure 3, the meaning and valid area of the figures are specified in detail.
Point 4: What is Eq. (14) used for? According to my understanding, in the paper, the run-up limit of edge waves is determined by Eq. (15), the gradient of the wave surface height, and Eq. (14) is not used and explained in the following texts. In my mind, the run-up limit should be determined by Eq. (14) rather than Eq. (15). Please give some justifications.
Response 4: As the bed level line is tangential to the water surface curve, we apply the following two equations Eqs .(14) and (15). The above sentence was described in line 153, and the meaning of the symbol was supplemented when the equation was developed.
Point 5: Eq. (17), what is e? The meaning of notation should be offered.
Response 5: The meaning of the symbol was supplemented when the equation was developed, see line 161.
Point 6: What is alpha_max? The meaning should be offered.
Response 6: When the water surface amplitude at the shoreline α is known, the wave number k is decided, if maximum run-up develops, or vice versa. The water surface maximum amplitude at the shoreline is . The meaning of the symbol was supplemented when the equation was developed, see line 160.
Point 7: Line 164, how is the value ‘0.214’ obtained?
Response 7: The value of 0.214 is the universal coefficient calculated using Eqs. (25) to (27), and condition of water surface maximum amplitude at the shoreline is 1.43m.
Point 8: The writing and language should be improved. For exmaple,
(1) Line 39, ‘have’ should be ‘has’;
(2) Line 95, ‘Figure 1(a) of Figure 1(b)’ may be ‘Figure 1(a) and 1(b)’
(3) Line 173, ‘ Two example cases are used to examine could demonstrated ... ‘. Error in grammar.
(4) More ....
Response 8:
(1): The verb “has” is grammatically correct. It has been corrected and specified on line 48 of the main text.
(2): The English description method of line 95 has been corrected. The sentences were explained using the two figures[Figure 1(a) and 1(b)] as examples, and the English grammar and fluency have been thoroughly revised once again and supplemented as shown in line 106.
(3): I revised the sentence and narrative method in accordance with English grammar. Thank you for your precise examination.

Reviewer 2 Report
11. Abstract has not yet demonstrated the methodology for this study, the significance and scholarly contributions of this study should be provided.
22. Introduction:
- - Literature reviews need to be written more concisely and thoroughly to highlight research gaps. The novelty of the study needs to be clarified.
- - The last paragraph must provide the content presented in the following sections.
33. Please provide clearly the tool to calculate the mass transport rate.
44. How is the reliability and error of the calculated model in this study evaluated?
55. The data depicted in Figure 1, Figure 2, Figure 3, Figure 4, and Figure 5 need to be verified. If references from other studies should be cited appropriately.
66. The mathematical equations presented in Finding Maximum Run-up Limit should provide adequate references with adequate citations.
77. The results of the calculations in the Field Scales of Alongshore Mass Transport need to be discussed more thoroughly. Comparative reviews with recent related studies should be provided to clarify new findings in this study.
88. Conclusions
- - The content in this section should not contain literature reviews and results in representations, as shown in paragraph “Existing theories on the alongshore wave-induced current driven by radiation-stress gradients of obliquely-approaching of short-period waves describe the underwater flow pattern [30], see Figure. 8”.
- - Need to be rewritten to clarify the results achieved, and need to provide limitations of the study as well as the direction in future work.
Author Response
Point 1: Abstract has not yet demonstrated the methodology for this study, the significance and scholarly contributions of this study should be provided.
Response 1: The academic significance emphasized in this paper was abbreviated and emphasized in the summary section. Thank you for your in-depth review comments and suggestions.
Point 2: iterature reviews need to be written more concisely and thoroughly to highlight research gaps. The novelty of the study needs to be clarified.
Response 2: Section 1(introduction) was revised and supplemented to clarify the necessity and purpose of the paper. Please consider the review.
Point 3: The last paragraph must provide the content presented in the following sections.
Response 3: In the last paragraph of the introduction, the contents of each session were mentioned so that the research procedure of this thesis could be included, and our research focus, applications, and scope were presented. Please consider the review.
Point 4: Please provide clearly the tool to calculate the mass transport rate.
Response 4: The clear meaning of each symbol introducing mass transport were presented in line 173.
Point 5: How is the reliability and error of the calculated model in this study evaluated.
Point 6: The data depicted in Figure 1, Figure 2, Figure 3, Figure 4, and Figure 5 need to be verified. If references from other studies should be cited appropriately.
Point 7: The mathematical equations presented in Finding Maximum Run-up Limit should provide adequate references with adequate citations.
Point 8: The results of the calculations in the Field Scales of Alongshore Mass Transport need to be discussed more thoroughly. Comparative reviews with recent related studies should be provided to clarify new findings in this study.
Response 5~8: The δ introduced in this paper is defined as a sufficiently validated coefficient by acting as a universal coefficient even under the conditions tested by reducing dx, dy, and dt, if the edge waves satify the maximum run-up condition. However, if the edge waves are lower than the maximum run-up condition, the integration coefficient of δ becomes smaller.
This paper is meaningful in introducing the maximum runup due to edge-waves and developing the quantitative calculation through the proposed formula.
Each figure was revised and presented to convey a clear meaning. Figure 8 was specially modified to emphasize the significance of this study by citing existing literature.
Point 9: The content in this section should not contain literature reviews and results in representations, as shown in paragraph “Existing theories on the alongshore wave-induced current driven by radiation-stress gradients of obliquely-approaching of short-period waves describe the underwater flow pattern [30], see Figure. 8”.
Response 9: The unnecessary literature review in the conclusion part has been corrected and it has been properly rewritten in the introduction part.
Point 10: Need to be rewritten to clarify the results achieved, and need to provide limitations of the study as well as the direction in future work.
Response 10: This paper is based on the ideal fluid assumption. However, the real field of the coast involves a variety of phenomena, such as bottom friction, penetration through sand or gravel, or the formation of a bottom boundary layer, and sediment transport involves many more factors on top of them. So that edge-waves with different mechanisms are involved. The possibility of existence and the need for a mass transport calculation method are described later in the conclusion section.

Round 2
Reviewer 1 Report
The manuscript may be suitable for publication
Author Response
Thank you for your comment
This manuscript is a resubmission of an earlier submission. The following is a list of the peer review reports and author responses from that submission.